# Integration of Soil Electrical Conductivity and Indices Obtained through Satellite Imagery for Differential Management of Pasture Fertilization

**João Serrano** [1,*], **Shakib Shahidian** [1], **José Marques da Silva** [1,2], **Luís Paixão** [2], **José Calado** [3] **and Mário de Carvalho** [3]

1   ICAAM, Departamento de Engenharia Rural, Escola de Ciências e Tecnologia, Universidade de Évora, P.O. Box 94, Évora 7002-554, Portugal; shakib@uevora.pt (S.S.); jmsilva@uevora.pt (J.M.d.S.)
2   Agroinsider Lda. (spin-off da Universidade de Évora), PITE, R. Circular Norte, NERE, Sala 18, 7005-841 Évora, Portugal; lgpaixao@gmail.com
3   ICAAM, Departamento de Fitotecnia, Escola de Ciências e Tecnologia, Universidade de Évora, P.O. Box 94, Évora 7002-554, Portugal; jcalado@uevora.pt (J.C.); mjc@uevora.pt (M.d.C.)
*   Correspondence: jmrs@uevora.pt; Tel.: +351-266-760-800

**Abstract:** Dryland pastures in the Alentejo region, located in the south of Portugal, normally occupy soils that have low fertility but, simultaneously, important spatial variability. Rational application of fertilizers requires knowledge of spatial variability of soil characteristics and crop response, which reinforces the interest of technologies that facilitates the identification of homogeneous management zones (HMZ). In this work, a pasture field of about 25 ha, integrated in the *Montado* mixed ecosystem (agro-silvo-pastoral), was monitored. Surveys of apparent soil electrical conductivity ($EC_a$) were carried out in November 2017 and October 2018 with a Veris 2000 XA contact sensor. A total of 24 sampling points ($30 \times 30$ m) were established in tree-free zones to allow readings of normalized difference vegetation index (NDVI) and normalized difference water index (NDWI). Historical time series of these indices were obtained from satellite imagery (Sentinel-2) in winter and spring 2017 and 2018. Three zones with different potential productivity were defined based on the results obtained in terms of spatial variability and temporal stability of the measured parameters. These are the basis for the elaboration of differentiated prescription maps of fertilizers with variable application rate technology, taking into account the variability of soil characteristics and pasture development, contributing to the sustainability of this ecosystem.

**Keywords:** pastures; spatial variability; temporal stability; $EC_a$; NDVI; NDWI; management zones; decision making

## 1. Introduction

Dryland pastures are the principal source of animal feed in extensive animal production systems in the Alentejo region of south of Portugal. These occupy soils that have low fertility and, simultaneously, important spatial variability in key parameters. The characteristic undulating relief, in association with intensive land use management (cereal monoculture) during decades, has originated mechanisms of erosion and soil transport, leading to degraded, shallow and stony soils, with low organic matter and nutrient content and a tendency to become acidic, all limiting factors for productivity [1]. Even productive and intensively managed grasslands comprise a wide range of vegetation types, and there is still uncertainty about the extent to which we can measure individual properties of that vegetation needed for application of automated, reliable, and time-saving management techniques. In addition to different vegetation types, the annual variations of floristic composition and vegetation dynamics

introduce significant variability and uncertainty into standardized sensing techniques on permanent grassland. Further, the situation on grassland becomes more complicated when grazing animals are involved. The grazing animals create specific spatial patterns of sward biomass that change throughout utilization with considerable effects on the spatial heterogeneity of the grassland field [2]. In the case of pastures under tree cover such as the *Montado* of cork oak (*Quercus suber* L.) or of holm oak (*Quercus ilex ssp. rotundifolia Lam.*) the effect of trees on pasture is significant [3] and a direct consequence of the extent to which these trees modify the microclimate and soil properties [4].

The first step for implementing strategies for management of this variability, which culminate in the differential application of production factors, namely the application of fertilizers with variable rate technology (VRT), requires the determination of the spatial and temporal patterns of the main soil properties and crop response. Therefore, assessing variability is the first critical step and a necessary condition in precision agriculture [5]. The intermediate step should lead to the identification and demarcation of areas with similar characteristics (soil and/or crop development), known as homogeneous management zones (HMZ) to implement site-specific management strategies [6,7].

Several studies have shown the practical interest and the potential of soil $EC_a$ monitoring for implementing smart sampling, designing and establishing HMZ, and elaborating prescription fertilizer maps [6,8,9]. This potential results from the fact that this parameter integrates the main properties affecting crop productivity, namely, texture, soil moisture, organic matter content, and soil cation exchange capacity [7–9]. Sensors to measure soil $EC_a$ in the field are of two types: Contact or non-contact. The contact sensor (e.g., "Veris 2000 XA" model) uses electrodes in the shape of coulters that make contact with the soil to measure the electrical resistivity. The non-contact sensor is non-invasive and works on the principle of electromagnetic induction (e.g., "EM38", "GEM-2", or "DUALEM 1 S") [6]. These sensors can be carried on mobile platforms mounted on a tractor or on an all-terrain vehicle, allowing quick $EC_a$ surveys and providing large amounts of information on various physical soil properties [5]. However, these geophysical methods have a cost, which varies depending on the area to be realized and may not be compatible with the low profit margin associated with extensive livestock production. Simultaneously, $EC_a$ measurements are generally affected by more than one agronomic soil characteristic and, consequently, obtaining accurate information about one property, by using only one sensing technique, is extremely difficult [5]. In addition, since spatial patterns in $EC_a$ measurements are affected by seasonal effects (e.g., weather conditions), single $EC_a$ maps will have limited interest in supporting decision making [7].

Recently, to obtain a more comprehensive representation of the surveyed area and separate more easily the different effects, a new approach for soil and vegetation sensing based on combining several sensing techniques (sensor fusion system), has been developed [5]. In this study, $EC_a$ measurements were associated with information obtained by remote sensing. The easy access to satellite images and historical series of various indices, including NDVI and NDWI, provide information with a very interesting spatial ("10 × 10 m" in NDVI and "20 × 20 m" in NDWI) and temporal resolution (five days) [10]. The NDVI is based on reflectance at the near infrared and red regions, which is strongly related to the vegetation density [11] or to the chlorophyll content, and thus, with the plant growth [12], being the most widely used vegetation index for definition of HMZ [13–15]. On the other hand, NDWI, because it incorporates a short-wave infrared band (SWIR), is sensitive to changes in liquid water, having been initially proposed for describing the status of the vegetation water content [16,17]. The NDWI responds to changes in both the water content and spongy mesophyll in vegetation [18].

The aim of this paper is to demonstrate the interest of associating expedited soil $EC_a$ survey with monitoring of time series of indices obtained from satellite imagery in order to define and validate HMZ (Figure 1). These techniques are used to evaluate the spatial variability and temporal stability of the measured parameters, which are fundamental aspects to support decision making regarding the elaboration of differentiated fertilizer prescription maps.

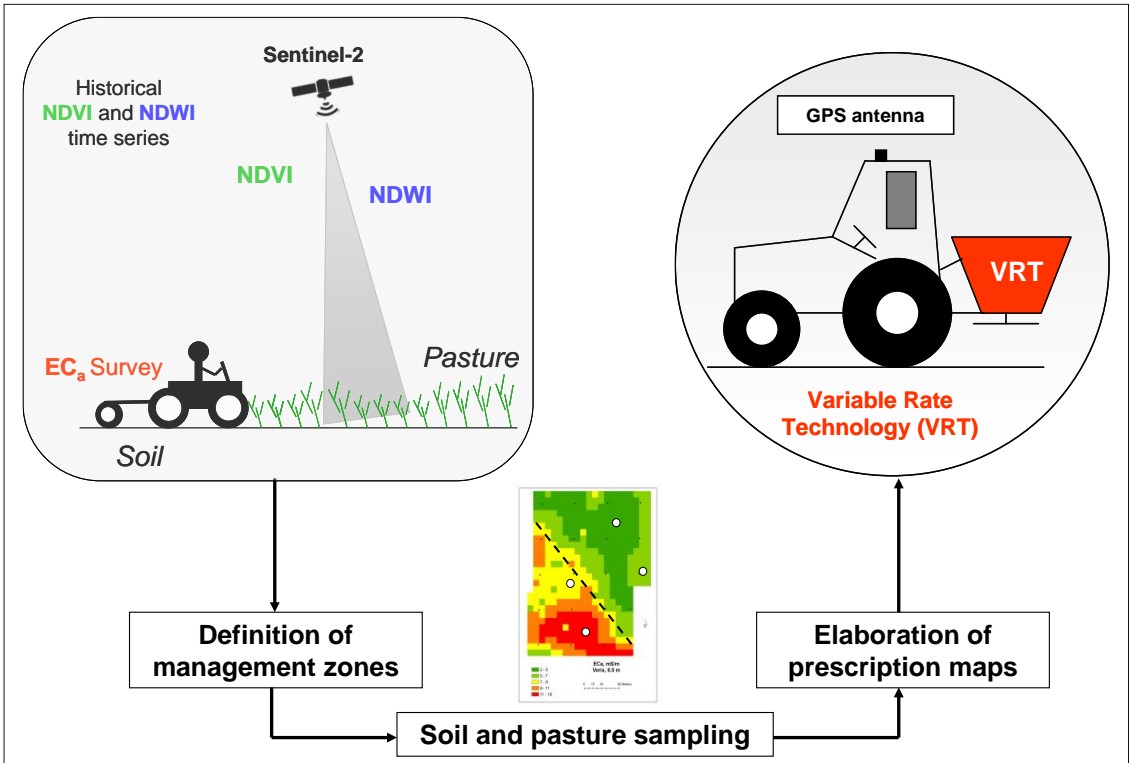

**Figure 1.** Schematic representation of the steps proposed in this work for establishing homogeneous management zones.

## 2. Materials and Methods

### 2.1. Experimental Field

The area of the study, a paddock, with 25 ha (Figure 2a), is located in the south of Portugal (coordinates 38°32.1′ N; 7°59.8′ W). This is a field of *Quercus ilex ssp. rotundifolia Lam.*, with a tree density of approximately 8–10 trees ha$^{-1}$, cultivated for more than 30 years with bio-diverse permanent pastures (grasses, legumes, and others) and used for sheep grazing. The dominating soil type is Cambisol with a granite origin [19]. Cambisols are characterized by slight or moderate weathering of parent material and by absence of appreciable quantities of illuviated clay, organic matter, aluminium, and/or iron compounds. These acid Cambisols are not very fertile and are mainly used for mixed agro-silvo-pastoral systems.

This work is part of a project to support farmers' decision making ("INNOACE") based on satellite imagery. It is therefore an agricultural field whose area is physically delimited by fences and where the animals graze. This is a real farmer's field and therefore it was decided to use it as is, in order to verify the resilience of the methodology under real field conditions.

Twenty-four georeferenced sampling areas were established (each with an area of 900 m$^2$, corresponding to "30 × 30 m" Sentinel-2 pixels). These are located in tree-free zones (Figure 2a) in order to allow readings of indices obtained from satellite images without interference from tree vegetation.

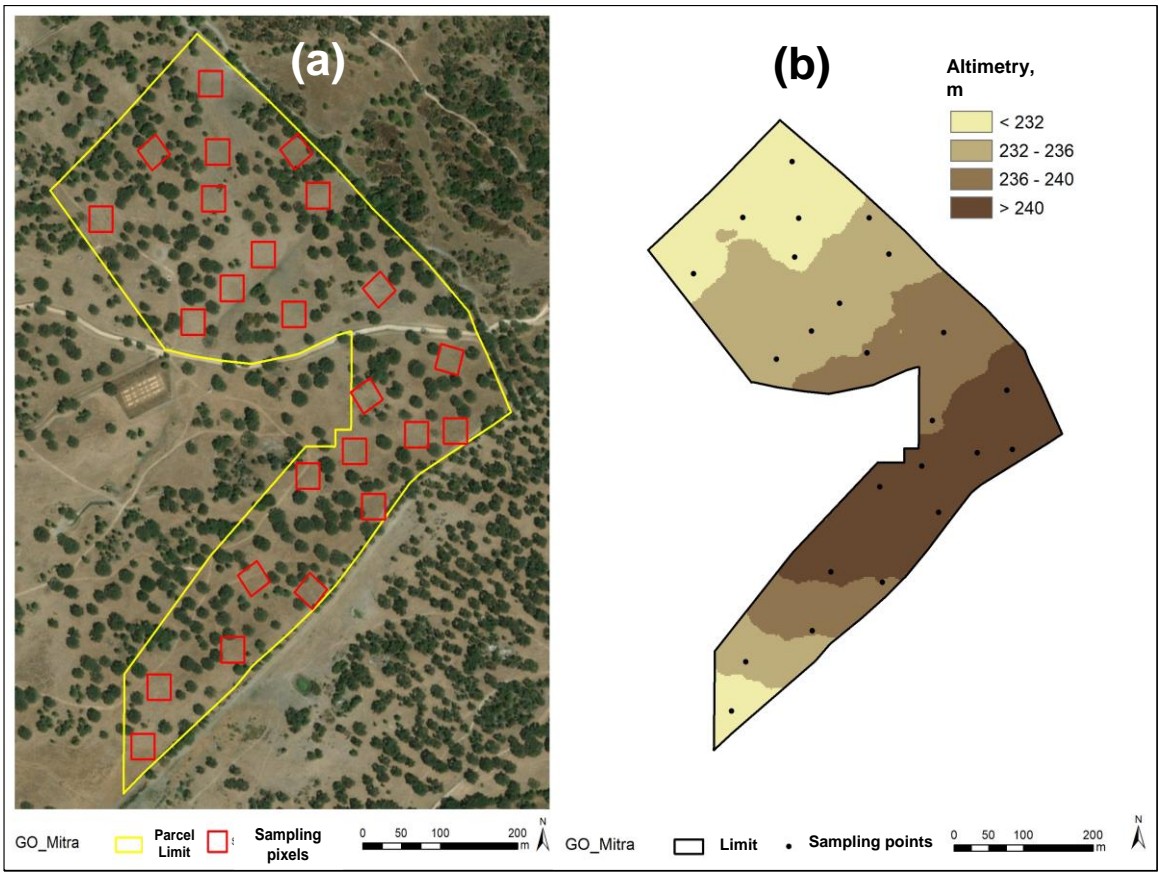

**Figure 2.** Limit of the experimental field with approximate location of the 24 sampling areas (**a**) and respective altimetry map (**b**).

### 2.2. Digital Elevation Model (Altimetry)

A topographic survey of the area was carried out in November 2017 using a real time kinematic (RTK) GPS instrument ("Trimble RTK/PP-4700 GPS", manufactured by TRIMBLE Navigation Limited, Sunyvale, CA, USA). The elevation data were recorded in the field with an all-terrain vehicle that was driven along transects in the field about 10 m apart. The digital surface elevation model (altimetry, Figure 2b) was created using the linear interpolation TIN tool from ArcGIS 9.3 and converted to a grid surface with a 1 m grid resolution.

### 2.3. Soil $EC_a$ Measurements

A Veris 2000 XA contact-type sensor (Veris Technologies, Salina, KS, USA) was used to measure $EC_a$ in November 2017 and October 2018 at 0–0.30 m depth. The sensor, equipped with a global positioning system (GPS) antenna, was pulled by an all-terrain vehicle at an average speed of 2.0 m s$^{-1}$ and successive passages spaced 10 m were made across the field. The $EC_a$ measurements were registered continuously every second, so the spatial resolution was a 2 by 10 m grid. After measuring $EC_a$, the soil moisture content (SMC) variability was assessed. Composite soil samples (comprised of nine subsamples) were collected in sampling areas at 0–0.30 m depth using a gouge auger and a hammer. Soil samples were transported to the lab in metallic boxes, weighed, and then dried at 105 °C for 24 h; once cooled they were weighed again, to establish soil moisture content (SMC). The volumetric SMC was then obtained through multiplying the values by the bulk density.

## 2.4. Soil Sample Collection and Analysis

Composite soil samples (comprised of nine subsamples) were collected in November 2017, in the 24 sampling areas. These soil samples were inserted in plastic bags, air-dried, and analyzed for particle-size distribution (texture: Sand, silt, and clay content) using a sedimentographer ("Sedigraph 5100", manufactured by Micromeritics), after passing the fine components through a 2 mm sieve. The fine soil (fraction with diameter < 2 mm) was characterized in terms of pH, organic matter (OM), cationic exchange capacity (CEC), phosphorus ($P_2O_5$), potassium ($K_2O$), magnesium (Mg), and manganese (Mn). These fine components were also analyzed for pH in 1:2.5 (soil:water) suspension, using the potentiometric method. The OM was measured by combustion and $CO_2$ measurement, using an infrared detection cell. The CEC was measured by the neutral ammonium acetate method. The $P_2O_5$ and $K_2O$ were extracted by the Egner–Riehm method, and $P_2O_5$ was measured using colorimetric method, while $K_2O$ content was measured with a flame photometer. The Mg and Mn were measured using atomic absorption spectrometry.

## 2.5. Pasture Sample Collection and Analysis

Pasture sample collection was carried out on 10 May 2018, at the peak of pasture production. Composite pasture samples were collected at nine representative locations (each with 0.25 $m^2$ area) within each selected sampling area. At each sampling point, pasture was cut at 1 to 2 cm above ground. It was then (i) weighted immediately to obtain fresh mass (green matter, GM, kg·ha$^{-1}$); (ii) dehydrated (72 h at 65 °C); and (iii) weighted again, in order to obtain pasture dry matter yield (DM, kg·ha$^{-1}$). The dehydrated samples were analyzed in order to determine the content of crude protein (CP) and neutral detergent fiber (NDF) according to standard methods (both in % of DM) [20].

## 2.6. Vegetation Multispectral Measurements by Remote Sensing

Through the electronic platform "http://agromap.agroinsider360.com" from the "AgroInsider" enterprise (a spin-off from the University of Évora), Sentinel-2 band 4 (B4; 10 m spatial resolution; 665 nm), band 8 (B8; 10 m spatial resolution; 842 nm), band 8 A (B8 A; 20 m spatial resolution; 865 nm), and band 11 (B11; 20 m spatial resolution; 1610 nm), atmospherically corrected imagery, were extracted from Copernicus data hub and used to calculate NDVI (Equation (1)) and NDWI (Equation (2)). These data (Sentinel-2 optical images) were obtained for the 24 sampling areas. NDVI of sampling area resulted from the average of the nine 10 × 10 m pixels that constitute this area and NDWI resulted from reading the 20 × 20 m pixels that contain the centre point of the sampling area. In order to make the reconstruction of historical NDVI and NDWI trends, NDVI and NDWI time series records from December 2016 to June 2017 and from December 2017 to June 2018 were retrieved. A preliminary processing was carried out on these records to remove outliers due to the presence of clouds. Only the images without presence of clouds were used in the analysis. In the period under consideration there were 30 available dates, 12 between December 2016 and June 2017 (6, 16, and 26/12; 15 and 25/1; 5, 15, and 25/4; 25/5; 4, 14, and 24/6) and 18 between December 2017 and June 2018 (1, 11, 16, 21, and 31/12; 15/1; 24/2; 21, 26, and 31/3; 25/4; 5, 10, and 15/5; 14, 19, 24, and 29/6).

$$NDVI = \frac{B8 - B4}{B8 + B4} \tag{1}$$

$$NDVI = \frac{B8A - B11}{B8A + B11} \tag{2}$$

## 2.7. Statistical Analysis of the Data

Descriptive statistics, including mean, standard deviation (SD), coefficient of variation (CV), and range, were determined for each dataset of the parameter measured in the sampling period.

The temporal stability of $EC_a$, NDVI, and NDWI was determined by calculating the CV at each sampling point over time (Equation (3)), using the method presented by Blackmore [21] and Xu et al. [22].

The average coefficient of variation ($\overline{CV}$) for all years (two years) and for all sampling points (24) was calculated (Equation (2)) to show the relative magnitudes of spatial and temporal variability; large values of $\overline{CV}$ indicate considerable spatial and temporal variation [23].

$$CV_i = \frac{\left(\dfrac{n\sum\limits_{t=1}^{t=n} y_{i_t}^2 - (\sum\limits_{t=1}^{t=n} y_{i_t})^2}{n(n-1)}\right)^{0.5}}{\overline{y_i}} \times 100 \tag{3}$$

where: $CV_i$ is the coefficient of variation over time at sampling point $i$; $y_{i_t}$ is the parameter at the sampling point $i$ at time $t$; $\overline{y_i}$ is the mean value at the $i$th sampling point over the $n$-year period and $n$ is the number of sampling years.

$$\overline{CV} = \frac{(\sum\limits_{i=1}^{m} CV_i)}{m} \tag{4}$$

where $m$ is the number of sampling points.

The two techniques described above quantify the spatial and temporal variation, which can be combined further into a single map of management classes, to be used for future decision making. These maps distinguish between different areas of the field based on their spatial and temporal characteristics. Each sampling point was represented by a coded value. The sampling points were classified by applying combinational logic statements to the spatial variation and temporal stability data sets, considering the following conditions: Condition 1 (relative value) identifies whether the point is above or below the average of all points for all the years; condition 2 (temporal stability) identifies the parameter stability at a particular point by comparing the CV to an arbitrary threshold (15% and 25%, stable and moderately stable, respectively). A point was considered to belong to a particular class if both conditions were true, and it was then assigned an arbitrary class code shown in brackets. The following five classes were established: (1) Greater than field mean value of parameter and stable (CV < 15%); (2) greater than field mean value of parameter and moderately stable (15% < CV < 25%); (3) smaller than field mean value of parameter and stable (CV < 15%); (4) smaller than field mean value of parameter and moderately stable (15% < CV < 25%); and (5) unstable (CV > 25%) [22].

From the mean values of $EC_a$ and the indices obtained by RS, for each sampling area and in the set of collections performed (two in the case of $EC_a$ and every five or ten days in the case of indices obtained by RS—after excluding those affected by cloudy days in the winter and spring periods of 2016/2017 and 2017/2018), three potential productive zones were established: High, medium, and low. The high potential productive zones presented high values (above average) in all evaluated parameters ($EC_a$, NDVI, and NDWI); the medium potential productive zones presented high values (above average) in the soil parameter ($EC_a$) or in the vegetation parameters (NDVI and NDWI); the low potential productive zones presented low values (below average) in all evaluated parameters ($EC_a$, NDVI, and NDWI). The definition of these three potential productive zones was validated by monitoring pasture productivity at the peak production period of spring 2018.

All maps were elaborated with the "ArcMap/Spatial Analyst" module of the ARCGIS 9.3 program [23], and the inverse distance weighting (IDW) interpolator was used.

An analysis of correlation between parameters was performed using the "20.0 SPSS package for Windows" (SPSS Inc., Chicago, IL, USA) with a significance level of 95% ($p < 0.05$). Correlations were established between: (i) The mean $EC_a$ of the two assessment dates and the corresponding mean SMC; (ii) the average NDVI and NDWI over the period considered (Autumn and Winter 2017 and 2018); (iii) NDVI, pasture green matter productivity (GM) and pasture crude protein (CP) on 10 May 2018.

## 3. Results

### 3.1. Spatial Variability

Table 1 summarizes the results of the descriptive statistical analyses (mean, standard deviation, coefficient of variation, and range) of soil characteristics in the top layer (0–0.3 m), pasture parameters and remote sensing indices, measured on different dates at the 24 sampling areas of the experimental field.

**Table 1.** Soil and pasture parameters and remote sensing indices, measured at the 24 sampling areas of the experimental field.

| Parameter (Date) | Mean | SD | CV | Range |
|---|---|---|---|---|
| Soil (November/2017) | | | | |
| $EC_a$. mS m$^{-1}$ | 2.3 | 1.3 | 55.4 | 0.8–5.5 |
| SMC, % | 9.4 | 1.8 | 18.6 | 7.4–12.5 |
| Sand, % | 78.4 | 4.0 | 5.0 | 71.5–84.6 |
| Silt, % | 11.2 | 2.2 | 19.9 | 7.4–15.3 |
| Clay, % | 10.4 | 1.8 | 17.3 | 7.2–13.9 |
| Organic matter, % | 1.5 | 0.3 | 21.5 | 0.9–2.1 |
| pH | 5.5 | 0.2 | 4.4 | 5.0–5.8 |
| $P_2O_5$, mg kg$^{-1}$ | 32.6 | 21.5 | 65.8 | 7.8–81.0 |
| $K_2O$, mg kg$^{-1}$ | 94.0 | 72.1 | 76.7 | 81.0–380.0 |
| Magnesium (Mg), mg kg$^{-1}$ | 57.7 | 25.3 | 43.8 | 15.0–120.0 |
| Manganese (Mn), mg kg$^{-1}$ | 33.0 | 18.0 | 54.4 | 15.0–87.0 |
| CEC, cmol kg$^{-1}$ | 10.8 | 2.8 | 26.4 | 5.2–17.9 |
| Remote Sensing (December 2016–June 2017 and December 2017–June 2018) | | | | |
| NDVI | 0.606 | 0.034 | 5.6 | 0.540–0.669 |
| NDWI | 0.276 | 0.042 | 15.2 | 0.202–0.349 |
| Pasture (May/2018) | | | | |
| Green matter, kg ha$^{-1}$ | 25,188 | 8776 | 34.8 | 7000–44,200 |
| Dry matter, kg ha$^{-1}$ | 3946 | 1053 | 26.7 | 1400–5700 |
| Crude protein, % | 12.1 | 1.9 | 15.6 | 8.9–15.5 |
| NDF, % | 51.4 | 3.6 | 7.0 | 45.7–58.0 |
| Soil (October/2018) | | | | |
| $EC_a$. mS m$^{-1}$ | 1.8 | 0.9 | 50.0 | 0.6–3.7 |
| SMC, % | 7.9 | 1.0 | 12.5 | 6.4–9.8 |

SD—Standard deviation; CV—Coefficient of variation; $EC_a$—Soil electrical conductivity; SMC—Soil moisture content; CEC—Cation exchange capacity; NDVI—Normalized difference vegetation index; NDWI—Normalized difference water index; NDF—Neutral detergent fiber.

One aspect that can be highlighted is the low $EC_a$ recorded in both performed surveys (on average around 2 mS m$^{-1}$). Figure 3 illustrates $EC_a$ maps of both assessment dates, with a systematic tendency towards higher values in the northern area and lower values in the southern area of the experimental field. It is also possible to verify, from 2017 to 2018, a slight decrease in the average $EC_a$ values in the experimental field.

Table 1 also highlights some of the main limitations of these soils: Acidic pH (around 5.5), poor in phosphorus ($P_2O_5$ on average 32.6 mg kg$^{-1}$), low OM content (1.5%), and poor CEC (10.8 cmol kg$^{-1}$). It is also evident the important spatial variability of soil parameters, reflected in the high CV, especially in $EC_a$ (50%–55%), in nutrient availability ($P_2O_5$, $K_2O$, Mg, Mn; 44%–77%), in CEC (26%), and OM (21.5%), and also present in terms of pasture productivity (green matter, GM, and dry matter, DM in kg ha$^{-1}$, with CV of the order of 25%–35%). Figures 4 and 5 show the spatial variability pattern of experimental field in terms of soil clay (Figure 4a), CEC (Figure 4b), OM (Figure 5a), and $P_2O_5$ (Figure 5b). The higher $EC_a$ in the northern part of the experimental field may reflect the higher soil clay content and CEC in these zones.

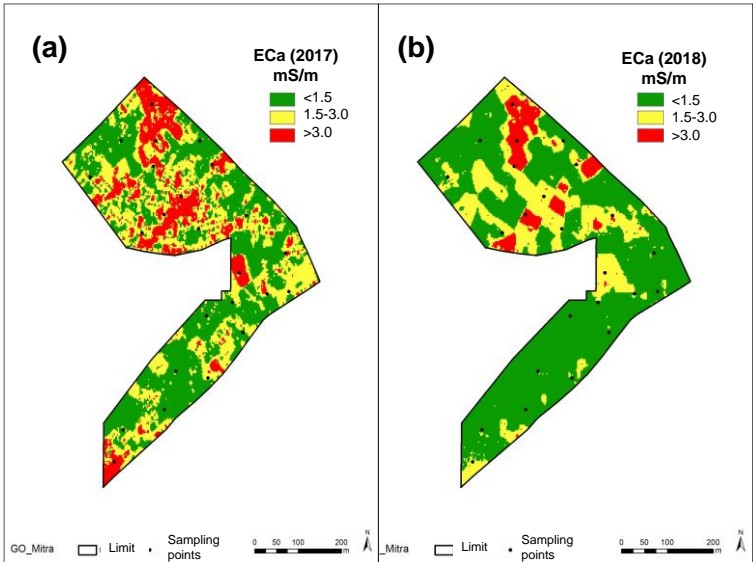

**Figure 3.** Soil electrical conductivity (EC$_a$) maps of experimental field in November 2017 (**a**) and October 2018 (**b**) at 0–0.30 m depth.

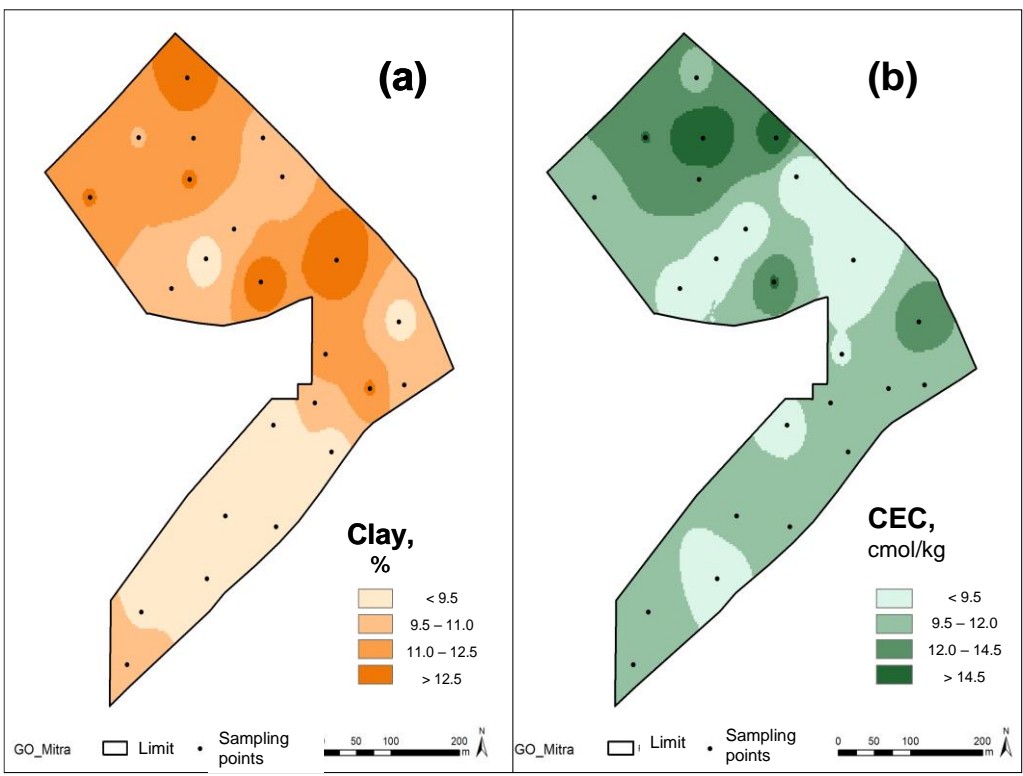

**Figure 4.** Soil clay (**a**) and cation exchange capacity (CEC) (**b**) maps of experimental field in November 2017 at 0–0.30 m depth.

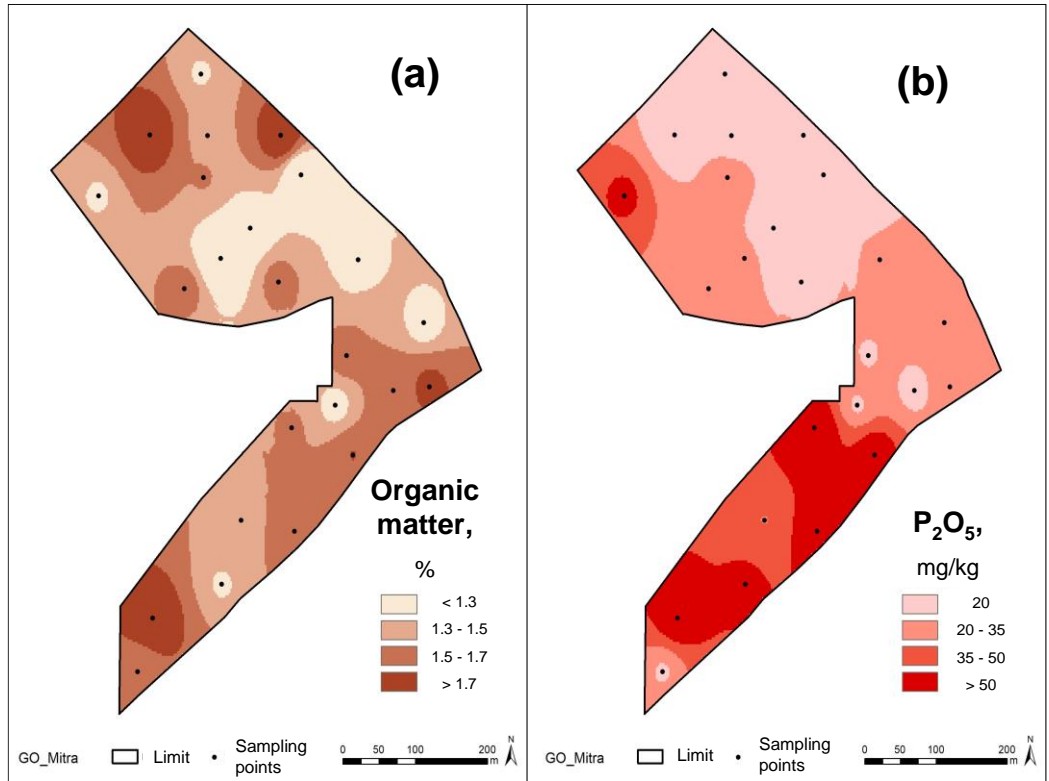

**Figure 5.** (**a**) Soil organic matter (OM) and phosphorus (P$_2$O$_5$) (**b**) maps of experimental field in November 2017 at 0–0.30 m depth.

Figure 6 shows NDVI and NDWI maps of the experimental field established from the mean of historical time series records of Sentinel-2 optical images captured between Winter and Spring of 2017 and 2018. These maps show the trend already identified with soil EC$_a$, with higher indices (and, therefore, greater vegetative vigor) in the northern zone and lower values in the southern zone of the experimental field.

### 3.2. Management Classes: Spatial Variability and Temporal Stability

These maps are a synopsis of the important features identified in the spatial variability and temporal stability, basic components of the new site-specific approach to crop management. The resulting maps of management classes of EC$_a$, NDVI, and NDWI are shown in Figure 7.

The temporal instability of EC$_a$ is evident in a significant area of the field (Figure 7a). In contrast, the map of management classes based on NDVI (Figure 7b) shows stability throughout the experimental field area (CV < 15%) and the map of management classes based on NDWI (Figure 7c) shows instability only in a small area (< 10% of the total) in the centre of the experimental field.

### 3.3. Site-Specific Management

The simultaneous integration of the three considered variables (EC$_a$, NDVI, and NDWI) allowed the elaboration of a proposal with three HMZ (Figure 8) based on the cumulative productive potential: (i) The high productive potential zone (with high values, above average, in all variables-EC$_a$, NDVI, and NDWI) represents about one third of the total area and is located in the northern part of experimental field; (ii) the medium productive potential zone (with high values, above average, in the soil parameter, EC$_a$, or in the vegetation parameters, NDVI and NDWI) represents about 40% of the total area of experimental and is relatively evenly distributed throughout the experimental field; (iii) the low productive potential zone which presented low values (below average) in all evaluated parameters

(EC$_a$, NDVI, and NDWI), occupies about a quarter of the total area and is located in the southern part of the experimental field.

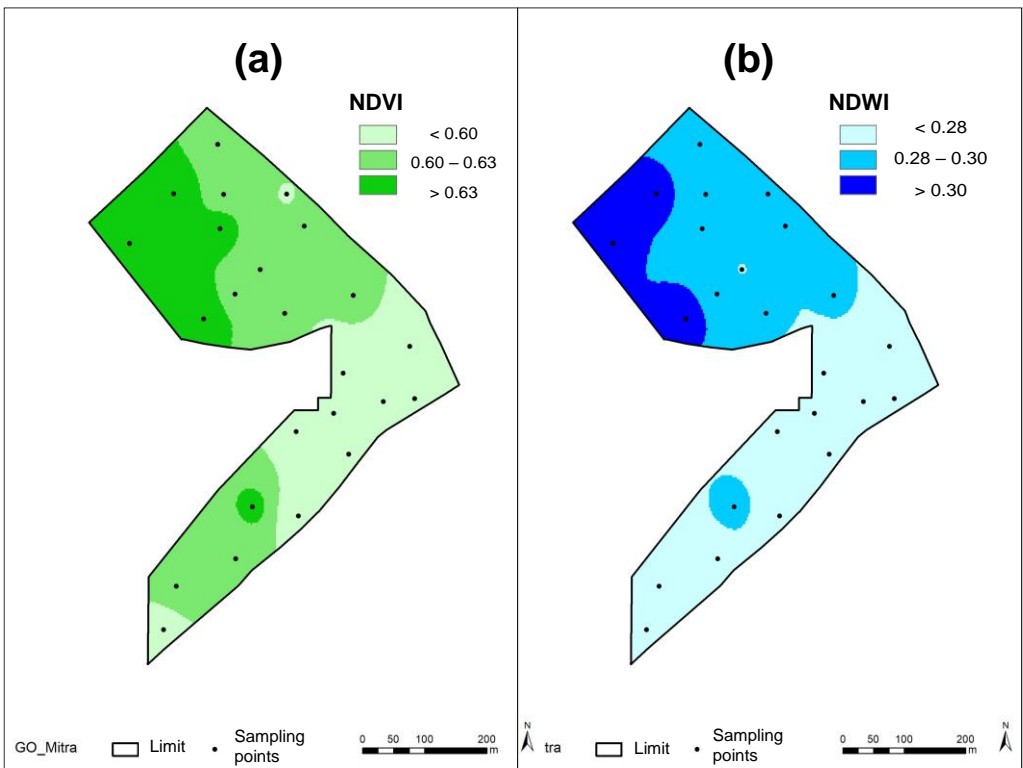

**Figure 6.** Normalized difference vegetation index (NDVI) (**a**) and normalized difference water index (NDWI) (**b**) maps of experimental field: Mean of historical time series records of Sentinel-2 optical images captured between Winter and Spring of 2017 and 2018.

Table 2 shows, for each zone, the mean and CV of soil characteristics, pasture parameters, and remote sensing indices. There is a gradient of increasing soil fertility (clay, SMC, and CEC), reflected in the EC$_a$ measured by the contact sensor, from the low potential zone to the high potential zone. However, the soil P$_2$O$_5$ and K$_2$O contents show an inverse trend, with lower values in the areas with the highest productive potential.

The definition of these three cumulative productive potential zones was validated by monitoring pasture productivity and quality at the peak production of spring 2018 (10 May 2018). Table 2 shows also a gradient of increasing pasture productivity (in terms of green matter, GM, and dry matter, DM) from the low to high potential zone.

## 4. Discussion

### 4.1. Spatial Variability

Soil is an extremely complex and highly variable medium and this poses significant challenges [15]. Spatial variability is the starting point for implementing differentiated management strategies that constitute the basis of precision agriculture [5]. The major aim of precision agriculture is to optimize crop management by addressing spatial variability, and thus optimize the use of farm inputs such as fertilizers and herbicides [7].

The spatial variability of EC$_a$ and other soil parameters in the experimental field shows the potential interest of this technique and the resulting information for decision support purposes in terms of definition of HMZ [8]. The slight decrease in the average EC$_a$ values in the experimental field from 2017 to 2018 may be associated with a decrease in the average soil moisture content (SMC) [24].

Figure 9 shows the significant correlation ($R^2 = 0.7088$) found in this study between the mean values of $EC_a$ and SMC on the two assessment dates for the 24 sampling areas.

In addition to the low SMC (8%–9%) as a consequence of autumn (2017 and 2018) with low rainfall in the region, the low $EC_a$ values reflect the texture of these soils (sandy loam, with only about 10% clay). Several studies demonstrate the significant and positive influence of SMC and soil clay content on $EC_a$ [7,13,15]. The inherent limitations of these soils (Cambisols: Mostly derived from granites, quartz-diorites, and sandstones) were also presented by Carvalho et al. in a regional study [25].

The spatial variability of soil parameters, reflected in the high CV, is caused by variations in climate, topography, parent material, vegetation, complex geological and pedological processes, and soil management practices [26]. This dynamic of soil nutrients and their spatial variability are characteristic of soils in southern Portugal, where undulating relief predominates (Figure 2b) [27,28], particularly in the *Montado* ecosystem, where trees and animals decisively affect the expression of this variability [2]. However, Table 1 shows that the CV for NDVI and NDWI (5.6 and 15.2, respectively) is much lower than those revealed by $EC_a$ measured by the Veris contact sensor and by soil parameters such as OM, clay, or nutrients (phosphorus, potassium, magnesium, or manganese). The same happens in the pasture quality parameters (CP, with CV of 15.6%; and NDF, with CV of 7.0%), which seems to indicate a significant relationship between these pasture quality parameters and the NDVI and NDWI indices, what is in line with other studies [10,13]. These results show that the pattern of variability detected at vegetation level by optical sensors from remote sensing does not reflect, therefore, the pattern and extent of soil nutrient variability. On the other hand, Figure 10 shows a strong correlation ($R^2 = 0.8432$) between NDVI and NDWI, demonstrating that both can be used to monitor pasture development status: The first mainly reflects chlorophyll content and the second water content, both indicators of pasture vegetative vigor [10].

**Table 2.** Soil and pasture parameters and remote sensing indices, in each management zone (MZ) of the experimental field.

| Parameter | MZ–Low Potential | | MZ–Medium Potential | | MZ–High Potential | |
|---|---|---|---|---|---|---|
| (Date) | Mean | CV | Mean | CV | Mean | CV |
| Soil (November/2017) | | | | | | |
| $EC_a$. mS m$^{-1}$ | 1.5 | 22.2% | 1.7 | 29.0% | 3.6 | 16.1% |
| SMC, % | 8.4 | 15.1% | 9.5 | 18.2% | 9.5 | 15.2% |
| Sand, % | 80.3 | 4.9% | 77.7 | 5.3% | 77.8 | 4.9% |
| Silt, % | 10.4 | 12.3% | 11.6 | 13.5% | 11.4 | 11.0% |
| Clay, % | 9.3 | 12.0% | 10.7 | 12.7% | 10.8 | 11.1% |
| Organic matter, % | 1.6 | 10.9% | 1.6 | 14.0% | 1.3 | 11.0% |
| pH | 5.6 | 4.5% | 5.4 | 4.2% | 5.5 | 4.2% |
| $P_2O_5$, mg kg$^{-1}$ | 53.2 | 36.8% | 30.1 | 45.3% | 21.6 | 34.1% |
| $K_2O$, mg kg$^{-1}$ | 93.7 | 41.3% | 106.8 | 53.7% | 78.3 | 36.5% |
| Magnesium (Mg), mg kg$^{-1}$ | 60.0 | 25.7% | 57.0 | 35.8% | 56.9 | 27.9% |
| Manganese (Mn), mg kg$^{-1}$ | 25.3 | 26.3% | 39.6 | 43.2% | 30.6 | 28.4% |
| CEC, cmolc kg$^{-1}$ | 10.2 | 14.0% | 10.5 | 18.3% | 11.4 | 15.7% |
| Remote Sensing (Winter and Spring 2017 and 2018) | | | | | | |
| NDVI | 0.573 | 4.3% | 0.618 | 4.8% | 0.617 | 5.1% |
| NDWI | 0.233 | 10.4% | 0.289 | 12.4% | 0.292 | 10.8% |
| Pasture (May/2018) | | | | | | |
| Green matter, kg ha$^{-1}$ | 23,283 | 25.5% | 24,463 | 29.7% | 26,910 | 18.4% |
| Dry matter, kg ha$^{-1}$ | 3633 | 20.1% | 3925 | 28.7% | 4150 | 18.0% |
| Crude protein, % | 12.0 | 13.9% | 11.4 | 14.0% | 12.7 | 10.5% |
| NDF, % | 49.9 | 4.1% | 51.6 | 7.5% | 52.3 | 4.7% |
| Soil (October/2018) | | | | | | |
| $EC_a$. mS m$^{-1}$ | 1.1 | 22.5% | 1.5 | 34.1% | 2.7 | 21.9% |
| SMC, % | 6.5 | 10.3% | 7.9 | 10.4% | 8.4 | 9.9% |

MZ—Management zones; SD—Standard deviation; CV—Coefficient of variation; $EC_a$ -Soil electrical conductivity; SMC—Soil moisture content; CEC—Cation exchange capacity; NDVI—Normalized difference vegetation index; NDWI—Normalized difference water index; NDF—Neutral detergent fiber.

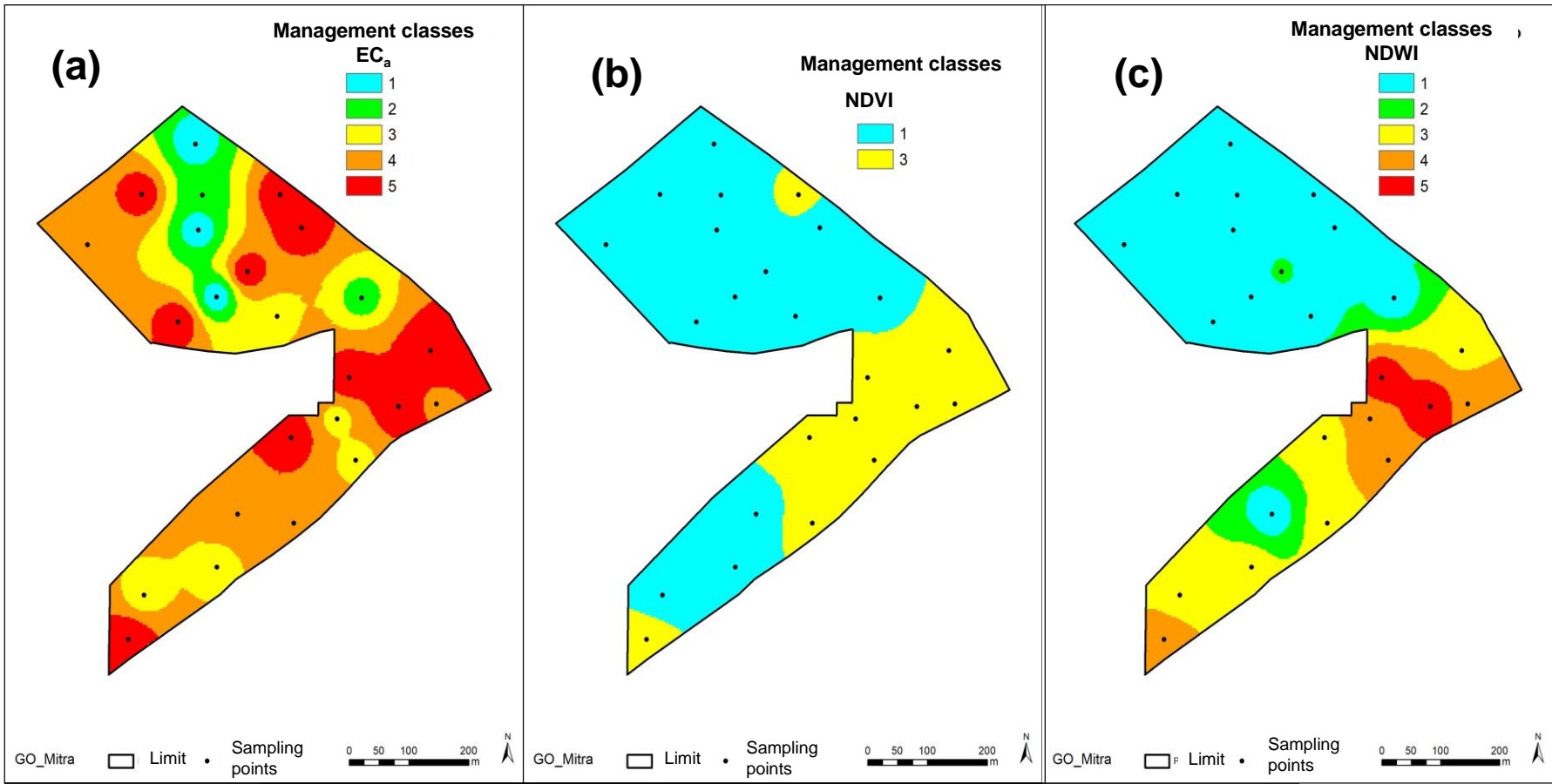

**Figure 7.** (**a**) Maps of management classes of the experimental field based on soil electrical conductivity (EC$_a$); (**b**) normalized difference vegetation index (NDVI); and (**c**) normalized difference water index (NDWI). Legend: 1—Greater than field mean value of parameter and stable; 2—Greater than field mean value of parameter and moderately stable; 3—Less than field mean value of parameter and stable; 4—Less than field mean value of parameter and moderately stable; 5—Unstable.

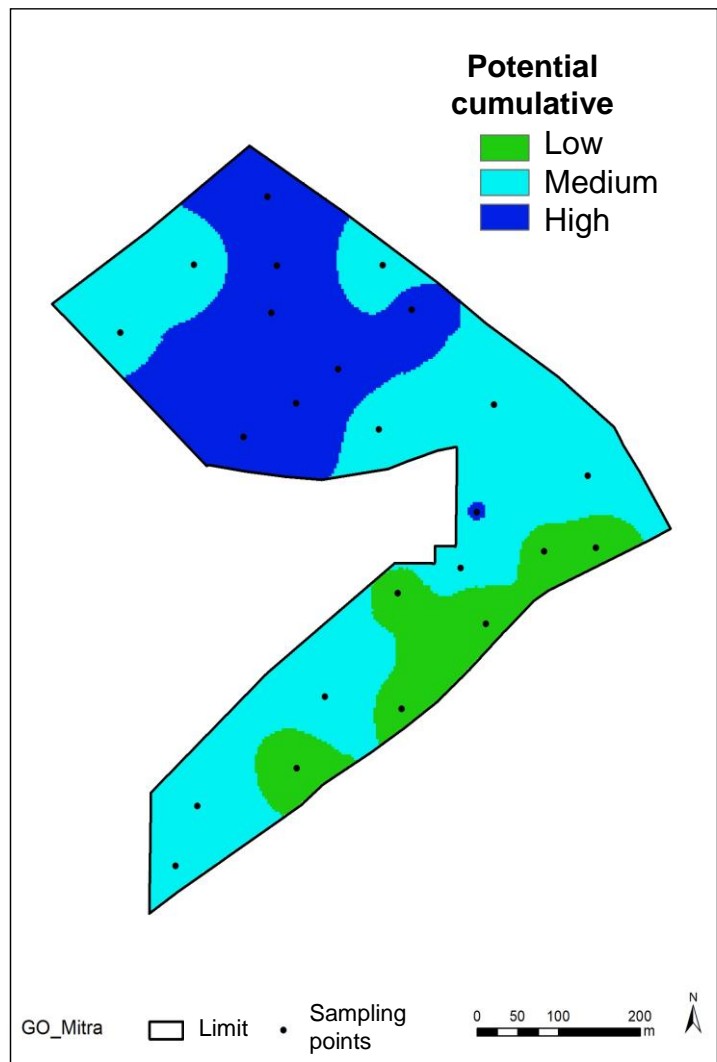

**Figure 8.** Potential cumulative map of experimental field.

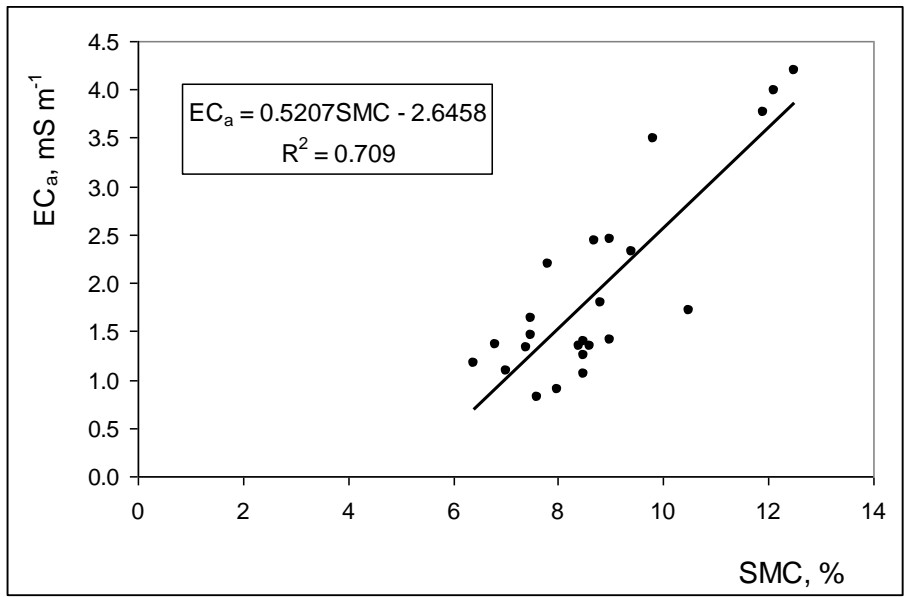

**Figure 9.** Relationship between mean soil electrical conductivity ($EC_a$) and mean soil moisture content (SMC) of experimental field at 0–0.30 m depth.

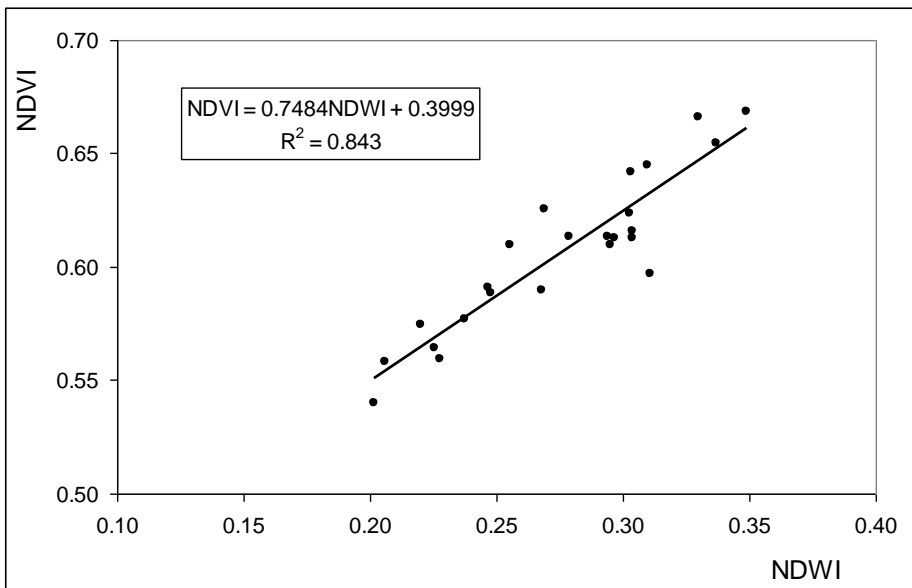

**Figure 10.** Relationship between mean values of normalized difference vegetation index (NDVI) and normalized difference water index (NDWI) of experimental field at 0–0.30: Historical time series records of Sentinel-2 optical images captured between Winter and Spring of 2017 and 2018.

### 4.2. Management Classes: Spatial Variability and Temporal Stability

The purpose of management class maps is to identify homogenous areas for research or application of soil amendments [21,28]. The temporal instability of $EC_a$ in a significant area of the field (Figure 7a) can make it difficult to define HMZs [9]. Peralta et al. [8] showed inconsistent relationships between ECa and soil characteristics, which can imply that the application of ECa in HMZ and site-specific management might not be feasible [22]. The patterns of $EC_a$ are influenced by a number of complex and mostly inter-related parameters, and are even affected by seasonal effects (e.g., weather conditions), therefore interpretation is not necessarily straightforward [7]. This is particularly relevant in areas of undulating relief (as is the case), with a significant effect on the formation of zones of higher humidity in the concave areas and lower humidity in the convex areas [27], which may disturb and influence other variables associated with soil characteristics (e.g., clay, CEC, pH, or OM [7]. According to Peralta et al. [8], surface topography (slope and aspect) plays an important role in the hydrological process, determines the level and location of run-off and infiltration, and the water-flow gradients, which result in salt transport.

On the other hand, the stability of the maps of management classes based on NDVI (Figure 7b) and on NDWI (Figure 7c) show the interest of remote sensing and vegetation indices to complement the information collected from $EC_a$ for delineation of HMZ, an aspect highlighted by Georgi et al. [7] and Nawar et al. [15]. Xu et al. [22] confirmed that high degree of temporal stability coupled with well-defined spatial patterns suggest that there is a real opportunity for site-specific management.

### 4.3. Site-Specific Management and Fertilizer Prescription Maps

The management classes can be defined as HMZ and can be used as a baseline for most farming decisions [15]. The soil $P_2O_5$ and $K_2O$ contents tend to have (Table 2) lower values in the areas with the highest productive potential, which should reflect the higher extractions resulting from the higher pasture productivity in these areas [28].

Effectiveness of the HMZ delineation can be assessed by several criteria. The measure of CV reduction of the MZ when compared to the within-field CV can be calculated [15].

Comparison of Tables 1 and 2 shows a clear decrease in CV in all the parameters evaluated after the definition of HMZ, especially in the zones of high and low cumulative potential. Zones of medium

cumulative potential have the highest CV values, which potentially reflects more heterogeneous zones as they exhibit contradictory behaviours in terms of the soil assessment parameter ($EC_a$) and pasture assessment parameters (NDVI and NDWI), revealing the greater complexity involved in defining this management zone.

Figure 11a shows the strong correlation between NDVI obtained by remote sensing and pasture productivity (GM) on 10 May 2018. The tendency of NDVI to saturate at high to pasture biomass (GM > 25,000 kg ha$^{-1}$) is also evident, an aspect already reported by other authors in high-coverage grassland (as is the case of this work, carried out during the peak spring pasture production) [29–33]. The saturation point is dependent on the pasture species, chlorophyll content, and the morphology of the plants [30].

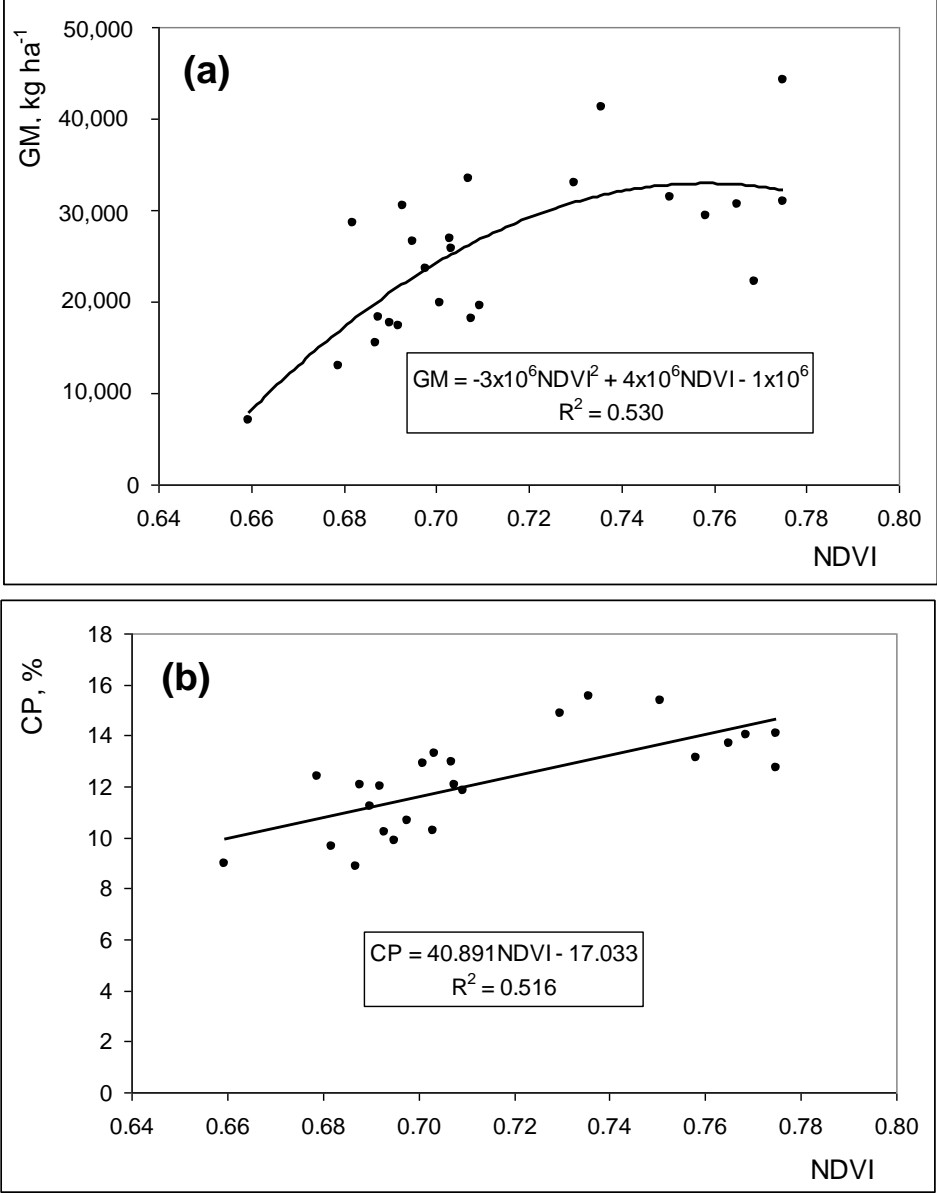

**Figure 11.** Relationship between normalized difference vegetation index (NDVI) and: (**a**) Pasture productivity (green matter, GM); (**b**) pasture quality (crude protein, CP) of experimental field in 10 May 2018.

Figure 11b shows the strong correlation between NDVI obtained by remote sensing and pasture quality (crude protein, CP) on 10 May 2018. Pullanagari et al. [34] and Albayrak [35] also found significant relationships between spectral measurements and pasture quality parameters, which can be attributed to the absorbance of visible radiance by the existing chlorophyll in green vegetation. These relationships reinforce the potential of using indices obtained by RS to establish HMZ and to manage *Montado* ecosystems, notably in the identification of animal supplementary feed requirements in the critical period of late spring and early summer [13].

The major aim of precision agriculture, and in particular the definition of HMZ is to optimize crop management by addressing spatial variability, and thus optimizing the use of farm inputs [7]. The recommended procedure for recovering degraded pastures in the Mediterranean region is to raise the soil phosphorus [36] to acceptable levels. The importance of phosphorus fertilizer application to increase pasture dry matter production was demonstrated by several researchers [28]. Sims et al. [37] identified four soil $P_2O_5$ categories for plant growth: Low ($0$–$25$ mg kg$^{-1}$), medium ($26$–$50$ mg kg$^{-1}$), optimum ($51$–$100$ mg kg$^{-1}$), and excessive ($> 100$ mg kg$^{-1}$). According to Efe Serrano [36] the application of phosphate chemical fertilizers on pastures should aim to achieve the reference values of $80$–$100$ mg $P_2O_5$ kg$^{-1}$ in order to favor the development of a bio-diverse flora, especially legumes. Table 3 shows the levels of $P_2O_5$ application rate defined based on classes of existing soil $P_2O_5$ concentration [38]. Figure 12 shows phosphorus prescription map ($P_2O_5$) proposed for the experimental field based on the references presented and on the three HMZ of the cumulative potential map (Figure 8). This differential fertilizer prescription map takes into account soil and pasture spatial variability. In the high cumulative potential zone, which has low soil $P_2O_5$ concentration ($21.6$ mg kg$^{-1}$; Table 2) [37], higher dosages of fertilizer are proposed ($80$ kg $P_2O_5$ ha$^{-1}$) to compensate for the higher plant extraction. In the medium cumulative potential zone, which has averages soil $P_2O_5$ concentration ($30.1$ mg kg$^{-1}$; Table 2) [37], a dosage of $60$ kg $P_2O_5$ ha$^{-1}$ is proposed, while in the low cumulative potential zone, which has optimum values of $P_2O_5$ ($53.2$ mg kg$^{-1}$; Table 2) [37], only a maintenance dosage of $30$ kg $P_2O_5$ ha$^{-1}$ is proposed.

An integrated view of all collected soil data (Table 1) shows that pH is relatively low throughout the experimental field ($5.5 \pm 0.2$; CV = $4.4\%$; Table 1). Given that the relative agronomic effectiveness of phosphorus (P) fertilizers and the availability of P in the soil environment in pastoral agriculture is governed by reactions in the soil matrix that are highly influenced by the pH [28], pH correction is justified before fertilizing with P. Carvalho et al. [15] added the need to improve the Mg/Mn ratio (average < 2 in this experimental field) to reduce the problems of Mn toxicity, which has long been recognized as the major limitation for the pasture and forage production in the Montado ecosystem on acid Cambisoils of the south of Portugal. These authors suggest that pH correction should be done, therefore, with the application of dolomitic limestone, which provides $CaCO_3$ but also Mg.

**Table 3.** Levels of phosphorus ($P_2O_5$) application rate based on classes of existing soil $P_2O_5$ concentration (adapted of Serrano et al. [38]).

| Soil $P_2O_5$ Concentration, mg kg$^{-1}$ | $P_2O_5$ Application Levels, kg ha$^{-1}$ |
|---|---|
| $[P_2O_5] < 30$ | 80 |
| $30 \leq [P_2O_5] < 50$ | 60 |
| $50 \leq [P_2O_5] < 80$ | 30 |
| $[P_2O_5] \geq 80$ | 0 |

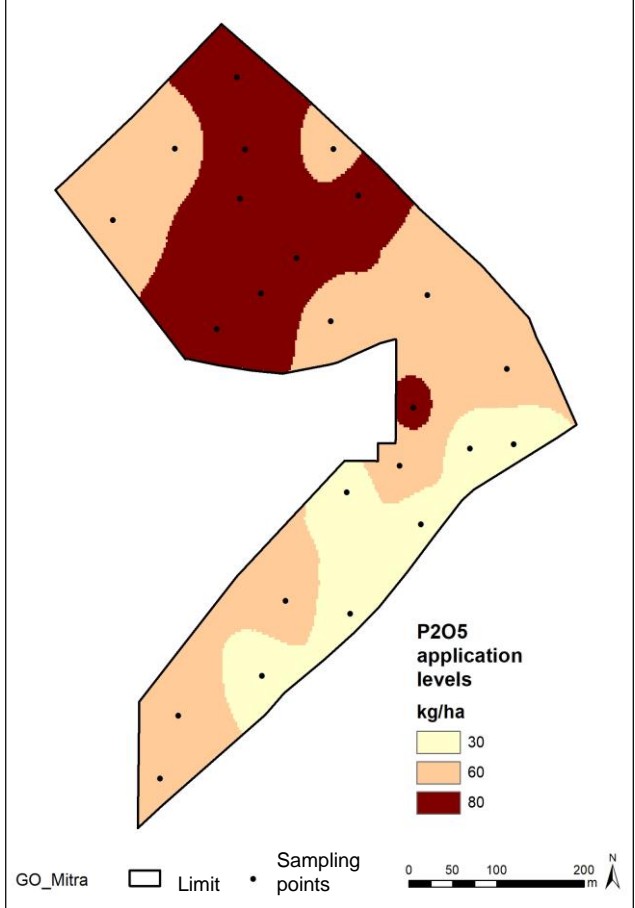

**Figure 12.** Phosphorus prescription map ($P_2O_5$ application levels) proposed for the experimental field based in three homogenous management zones (HMZ).

## 5. Conclusions

The dominant practice for farmers is to apply the same rate of fertilizer over whole fields and even whole farms. This practice leads to frequent over and under-application of fertilizers, a critical challenge to profitable crop production and long-term soil and environmental quality. Site-specific management is an attractive and intuitive approach to increasing the fertilizer use efficiency of agricultural systems by adjusting fertilizer rates to the soil and crop variability. An integrated approach, using spatial variability and temporal stability of $EC_a$ (measured by a contact sensor) and NDVI and NDWI (obtained by remote sensing), were used to delineate three management zones with differential productive potential, validated by soil and pasture sampling. The corresponding differential prescription map of phosphorus fertilizer was presented: Higher dosages were proposed in areas with high productive potential to compensate higher plant extraction of nutrients.

**Author Contributions:** Conceptualization, J.S., J.M.d.S., J.C. and M.d.C.; formal analysis, J.S., J.M.d.S., L.P. and M.d.C.; funding acquisition, J.S. and J.M.d.S.; investigation, J.S. and J.C.; methodology, J.S., S.S., L.P., J.M.d.S., J.C. and M.d.C.; project administration, J.S.; supervision, J.S. and M.d.C.; validation, J.S., S.S. and M.d.C.; visualization, J.S. and S.S.; writing —original draft preparation, J.S. and J.M.d.S.; writing—review and editing, S.S.

**Funding:** This work was funded by National Funds through FCT (Foundation for Science and Technology) under the Project UID/AGR/00115/2019, by the project INNOACE—Innovación abierta e inteligente en la EUROACE (Tarea 2.1.3) and by the projects PDR2020-101-030693 and PDR2020-101-031244 ("Programa 1.0.1-Grupos Operacionais").

**Acknowledgments:** The authors would like to express their sincere appreciation to Emanuel Carreira and Maria da Graça Machado of the Animal Production Department of the University of Évora for their technical support in, respectively, collection and analysis of data of pasture.

**Conflicts of Interest:** The authors declare no conflict of interest.

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
