# Peer review of "Integration of Soil Electrical Conductivity and Indices Obtained through Satellite Imagery for Differential Management of Pasture Fertilization"

_agriengineering, doi:10.3390/agriengineering1040041_

Round 1

Reviewer 1 Report

The proposed paper is interesting and worth consideration for publication, however some points must be addressed.

With regard to Soil ECa measurements, please explicitely describe data resolution: data were collected through 10 m distant passages, so the lateral resolution is 10 m; what about the resolution (i.e. the distance between two subsequent collected points) within each run: 2m  is it correct?

To scan the whole area with the Veris, at least 4 (but maybe 5) hours were needed: did you have variability on soil mosture or soil temperature during analysis, affecting results? How such variability/drift was eventually compensated? 

Why two passages were carryed out with Veris in such a short time (1 year)? ECa map in 2017 Is clearly affected by variability (noise?) (with respect to 2018 map): please discuss in a deeper why reasons behind such large change in soil variability.

5 management classes looks high for ECa: is that high number of classes confirmed in the two passages?

With regard to Vegetation multispectral measurements, further information must be provided with regard to number of available dates, accepted level of clouds, and variability. As I understand Figure 6 proposes average values, however it would be important to study and report also variability in VI indicies, in order understand the stability of the data set. 

Sampling areas were established with an area of 900 m2 corresponding to “30m × 30m” Sentinel-2 pixels: this is not clear. The used wavelengths (the ones implemented for vegetation indices) for the Sentinel 2 are 20 m not 30 m (B4 10 m; B8 10m; B8A 20 m and B11 20 m). SO please clarify this point. 

Avoid repetitions useless, e.g. Composite soil samples were collected at 0–0.30 m depth using a gouge auger and a hammer. 

Yield from previous years is always one of the most important factors to be considered for VRA. Why such information was not considered?

Therefore: how is yield improvement after application HMZ/VRA management?

In which date/dates was fertilization done? Did you ever fertilize the area before the beginning of the experiment?

Applied tools (ECa, soil sampling, data processes) have costs which are compensated by Precision Agriculture approaches. Indeed VRA can reduce costs and environmental impact: a quantification of this would be important to improve the quality of the paper.  

Author Response

“Integration of soil electrical conductivity and indices obtained through satellite imagery for differential management of pasture fertilization

“Special Issue "Selected Papers from 10th Iberian Agroengineering Congress"

Agriengineering-633636

___________________________________________________

Reviewer 1- Comments and Suggestions for Authors

The proposed paper is interesting and worth consideration for publication, however some points must be addressed.

R- The authors would like to thank the reviewers' comments and suggestions, which have greatly improved the final version of the article.

Comment 1

With regard to Soil ECa measurements, please explicitely describe data resolution: data were collected through 10 m distant passages, so the lateral resolution is 10 m; what about the resolution (i.e. the distance between two subsequent collected points) within each run: 2m  is it correct?

R- Yes, given that the sensor was programmed to record values at one second interval and the working speed was 2 m s-1 (as indicated in the text). As suggested by the reviewer, data spatial resolution (2 m by 10 m grid) was added to the text.

Comment 2

To scan the whole area with the Veris, at least 4 (but maybe 5) hours were needed: did you have variability on soil mosture or soil temperature during analysis, affecting results? How such variability/drift was eventually compensated? 

R- The survey was conducted on a November morning, between 8 and 12 noon, and at this time of year the outside temperature varies little (10-14°C). At the same time we’ve to state that soil temperature doesn’t affect the measurements of this soil resistivity equipment. Soil moisture was also measured (9.4 ± 1.8%) but considering the equipment’s soil scanning depth, the soil moisture values are not expected to change in 4 or 5 hours.

Comment 3

Why two passages were carryed out with Veris in such a short time (1 year)? ECa map in 2017 Is clearly affected by variability (noise?) (with respect to 2018 map): please discuss in a deeper why reasons behind such large change in soil variability.

R- The two Veris sensor passages were performed in two consecutive years (2017 and 2018) because it was intended to establish the ZHG for fertilizer application in 2019. In Precision Agriculture, it is essential to evaluate the spatial variability of soil and crops and the temporal stability of these spatial patterns. Soil spatial/temporal variation is apparent because the ECa values are very low in both maps (<5 mS/m). ECa values are very small, indicating in general, for all the field area, the presence of  sandy soils with low soil organic matter.

Comment 4

5 management classes looks high for ECa: is that high number of classes confirmed in the two passages?

R- Only 3 ECa classes were considered (<1.5; 1.5-3 and> 3 mS/m). The 5 management classes were obtained from two techniques described by various authors to quantify the spatial and temporal variation: condition 1 (relative value) identifies whether the point is above or below the average of all points for all the years; condition 2 (temporal stability) identifies the parameter stability at a particular point by comparing the CV to an arbitrary threshold (15% and 25%, stable and moderately stable respectively). In this particular case, the classes were obtained considering the two Veris sensor surveys.

Comment 5

With regard to Vegetation multispectral measurements, further information must be provided with regard to number of available dates, accepted level of clouds, and variability. As I understand Figure 6 proposes average values, however it would be important to study and report also variability in VI indices, in order understand the stability of the data set. 

R- Some information requested by the reviewer was already in the article:

(i) The variability of NDVI and NDWI indices (mean ± SD; CV; range) is presented in Table 1; and

(ii) Only the images without clouds were used in the analysis.

Information on the number of days available in the period under consideration (Dec 2016-Jun2017 and Dec 2017-June 2018) has now been added to the article: (i) Dec 2016 - Jun 2017 -12 available dates (6, 16 and 26/12; 15 and 25/1; 5, 15 and 25/4; 25/5; 4, 14 and 24/6); (ii) Dec 2017-Jun 2018 – 18 available dates (1, 11, 16, 21 and 31/12; 15/1; 24/2; 21, 26 and 31/3; 25/4; 5, 10 and 15/5; 14, 19, 24 and 29/6).

Comment 6

Sampling areas were established with an area of 900 m2 corresponding to “30m × 30m” Sentinel-2 pixels: this is not clear. The used wavelengths (the ones implemented for vegetation indices) for the Sentinel 2 are 20 m not 30 m (B4 10 m; B8 10m; B8A 20 m and B11 20 m). So please clarify this point. 

R- NDVI of sampling area resulted from the average of the nine 10 m × 10 m pixels that constitute this area; NDWI resulted from reading the 20 m × 20 m pixels that contains the center point of the sampling area. This text has been added in the new version of the article.

Comment 7

Avoid repetitions useless, e.g. Composite soil samples were collected at 0–0.30 m depth using a gouge auger and a hammer. 

R- As suggested by the reviewer, the duplicate text has been removed.

Comment 8

Yield from previous years is always one of the most important factors to be considered for VRA. Why such information was not considered? Therefore: how is yield improvement after application HMZ/VRA management?

R- Pasture productivity was measured in May 2018 and used in this paper to provide an example of spatial variability. However, unlike other crops (such as cereals, for example) whose harvesting is done by machines that may be equipped with electronic yield meters, pasture is grazed by animals in the field. Manual measurement of productivity (as done in this paper) is a lengthy process, involving a significant amount of manpower and is therefore impractical. In order to simplify the procedure and dvelop an expeditious and relatively inexpensive sampling method, we used indices (NDVI and NDWI) obtained by remote sensing and that once validated, can represent the spatial and temporal variability of pasture development. As for the question about the effect of applying HMZ/VRA, it is still not possible to answer because it is being applied only now (November 2019).

Comment 9

In which date/dates was fertilization done? Did you ever fertilize the area before the beginning of the experiment?

R- Soil acidity correction (with dolomitic limestone to reduce the negative effects of manganese toxicity) and phosphate fertilization are the recommended strategies for improving dryland pastures in the region. However, the profit margin of these extensive production systems is reduced due to low yields, and thus farmers choose not to make these applications. In this particular field, the last soil amendment was made in 2010 (2,000 kg ha-1) and the last fertilizer application (superphosphate 18% - 150 kg ha-1) was made in 2014. That is why we proposed this application. It will be implemented during November 2019.

Comment 10

Applied tools (ECa, soil sampling, data processes) have costs which are compensated by Precision Agriculture approaches. Indeed VRA can reduce costs and environmental impact: a quantification of this would be important to improve the quality of the paper. 

R- In this case, only after applying the fertilizer and evaluating its effect on pasture can we have an idea of the impact of this strategy (HMZ / VRA) on productivity. However, given the inter-annual variability of rainfall in this region, it may occur that autumn is dry and the whole strategy is compromised. What we can already mention is that the proposal aims to save fertilizer, taking into account the spatial variability of soil and pasture, applying larger quantities in the most productive areas (where P levels are lower) and smaller amounts in the less productive areas (and higher levels of P), which will naturally have an environmental impact.

Reviewer 2 Report

The article seems finished. Good language, I have a few questions about the whole experiment:

Location of experiment: Why chosen this field. It is representative for region? Why shape of experiment field is not regular. What was reason for plotting such an irregular shape? Is sampling location base on NDVI map? or what was the reason for their location? is it possible to attach a map showing the landform of terrain? 155 - the link doesn’t work. Please check correct of address. If it is necessary to login to take information please provide accurate information. 226 - What is the reason for low ECa recorded and in differences around south and north part of field? The imprecise authors provide the authors' data. For example, references 24 and 28 are Serrano J. instead of Serrano J.M. In references 5, 11 and 29, the authors give their full titles of capital letters instead of capital letters. figure 9-11 - R2 value suggests to round to 0.001 P2O5 in figure 5 should be given with a subscript

Author Response

“Integration of soil electrical conductivity and indices obtained through satellite imagery for differential management of pasture fertilization

Special Issue "Selected Papers from 10th Iberian AgroEngineering Congress"

Agriengineering-633636

___________________________________________________

Reviewer 2- Comments and Suggestions for Authors

The article seems finished. Good language, I have a few questions about the whole experiment:

R- The authors would like to thank the reviewers' comments and suggestions, which have greatly improved the final version of the article.

Comment 1: Location of experiment: Why chosen this field. It is representative for region? Why shape of experiment field is not regular. What was reason for plotting such an irregular shape?

R- This work is part of a project to support farmers' decision making (“INNOACE”) based on satellite imagery. It is therefore a farmland whose area is physically limited by fences and where the animals graze. This is a real farmer´s field and therefore it was decided to use it as is, in order to verify the resilience of the methodology under real field conditions. This information has been added to the text of the article.

Comment 2: Is sampling location base on NDVI map? or what was the reason for their location?

R- Sampling areas were located in tree-free zones (Figure 2(a)) in order to allow readings of indices obtained from satellite images (Sentinel-2 pixels) without interference from tree canopy and distributed evenly over the total area of the field.

Comment 3: Is it possible to attach a map showing the landform of terrain?

R- Relative field elevation (Altimetry map) is show in Figure 3b.

Comment 4: 155 - the link doesn’t work. Please check correct of address. If it is necessary to login to take information please provide accurate information.

R- The link was tested in different browsers and computers and was working well. We suggest to test again or ask for specialized help. If the same thing happens again please send an email to help@agroinsider.com in order to solve the problem.

Comment 5: 226 - What is the reason for low ECa recorded and in differences around south and north part of field?

R- In the “Discussion” section (lines 319-323) some possible reasons for low ECa values are presented:

“In addition to the low SMC (8-9%) as a consequence of autumn (2017 and 2018) with low rainfall in the region, the low ECa values reflect the texture of these soils (sandy loam, with only about 10% clay). Several studies demonstrate the significant and positive influence of SMC and soil clay content on ECa [7,13,15]. The inherent limitations of these soils (Cambisols: mostly derived from granites, quartz-diorites and sandstones) were also presented by Carvalho et al. in a regional study [25].”

The differences around south and north part of the field may reflect the higher soil clay content and CEC zones in the north part of the field (Figure 4, a and b). This hypothesis has been added to the text:

“The higher ECa in the northern part of the experimental field may reflect the higher soil clay content and CEC in these zones.”

Comment 6:  The imprecise authors provide the authors' data. For example, references 24 and 28 are Serrano J. instead of Serrano J.M. In references 5, 11 and 29, the authors give their full titles of capital letters instead of capital letters.

R- The references have been verified and corrected according to the reviewer's comment.

Comment 7: figure 9-11 - R2 value suggests to round to 0.001

R- Reviewer's suggestion was incorporated into the new version of the article.

Comment 8: P2O5 in figure 5 should be given with a subscript 

R- Reviewer's suggestion was incorporated into the new version of the article.

Round 2

Reviewer 1 Report

The proposed paper is an interesting research work which proposed integration of soil mapping techniques with satellite derived indices, to allow optimization of management of pasture fertilization.

The paper is well written and clear. Just some points have to be clarified before publication: 

1) what is the reason behind the choice of the homogenous zones?

The number of zones is typically based on variability or on expected profitability. 25 ha are enough to allow identification of 4 or 5 zones: why 3? Please support your choice and add comments for this ini the papers. 

2) Figure quality must be increased, using larger fonts

3) In the maps, the legends are not continuous (e.g. 9.7-12; 13-14 => it should be 9.7-12; 12-14 otherwise there are holes (not covered values))

4) zones are segmented with classes having different size (e.g. in the case of organic matter you use: 0.9-1.3; 1.4-1.5; 1.6-1.7; 1.8-2.1): such dishomogenous classes are just for representation puropose or were specifically adopted for some reason (in case please explain in the paper)

5) Authors say that "Figure 12 shows phosphorus prescription map proposed for the experimental field based on the references presented and on the three HMZ of the cumulative potential map (Figure 8)." This passage is fundamental and must be better clarified. I.e. authors have to better clarify if the adoption of Serrano values (30, 60, 80 kg/ha P2O5) are reasonable in this area: in other words the condition of the experimental area in this paper is comparable with that proposed by Serrano in his papers?

6) I miss a cost/benefits analysis in this paper: is the overall proposed procedure profitable at the end? are time and efforst reasonable with investments in proximal and remote sensing analyses?

Author Response

“Integration of soil electrical conductivity and indices obtained through satellite imagery for differential management of pasture fertilization

“Special Issue "Selected Papers from 10th Iberian Agroengineering Congress"

Agriengineering-633636

___________________________________________________

Reviewer 1- Comments and Suggestions for Authors (Round 2)

The proposed paper is an interesting research work which proposed integration of soil mapping techniques with satellite derived indices, to allow optimization of management of pasture fertilization.

 The paper is well written and clear. Just some points have to be clarified before publication: 

R- The authors would like to thank the reviewers' comments and suggestions, which have greatly improved the final version of the article.

Comment 1: what is the reason behind the choice of the homogenous zones? The number of zones is typically based on variability or on expected profitability. 25 ha are enough to allow identification of 4 or 5 zones: why 3? Please support your choice and add comments for this in the papers. 

R- The criterion for selection of the homogeneous zones and management classes in this work is merely based on the patterns of spatial variability and temporal stability, as explained in the methodology. The number of classes is 3, but in practice these represent in this field 4-5 areas with different potential (see Figures 8 and 12).  

Comment 2) Figure quality must be increased, using larger fonts

R- The suggestion was accepted.

Comment 3) In the maps, the legends are not continuous (e.g. 9.7-12; 13-14 => it should be 9.7-12; 12-14 otherwise there are holes (not covered values))

R- The reviewer is right. There were errors in the class boundaries, which were corrected.

Comment 4) zones are segmented with classes having different size (e.g. in the case of organic matter you use: 0.9-1.3; 1.4-1.5; 1.6-1.7; 1.8-2.1): such dishomogenous classes are just for representation purpose or were specifically adopted for some reason (in case please explain in the paper)

R- The reviewer is right. There were errors in the class boundaries, which were corrected.

Comment 5) Authors say that "Figure 12 shows phosphorus prescription map proposed for the experimental field based on the references presented and on the three HMZ of the cumulative potential map (Figure 8)." This passage is fundamental and must be better clarified. I.e. authors have to better clarify if the adoption of Serrano values (30, 60, 80 kg/ha P2O5) are reasonable in this area: in other words the condition of the experimental area in this paper is comparable with that proposed by Serrano in his papers?

R- Yes, the reference value of 80-100 mg P2O5 kg-1 as a threshold for promoting the development of a bio-diverse pasture, especially legumes, in typical soils in this region (Efe Serrano, 2006) was used in this study to define the classes of phosphate chemical fertilizers (30, 60 and 80 kg P2O5 ha-1).

It seems to us that this approach is clear in the text.                                

“This differential fertilizer prescription map takes into account soil and pasture spatial variability. In the high cumulative potential zone, which has low soil P2O5 concentration (21.6 mg kg-1; Table 2) [37], higher dosages of fertilizer are proposed (80 kg P2O5 ha-1) to compensate for the higher plant extraction. In the medium cumulative potential zone, which has averages soil P2O5 concentration (30.1 mg kg-1; Table 2) [37], a dosage of 60 kg P2O5 ha-1 is proposed, while in the low cumulative potential zone, which has optimum values of P2O5 (53.2 mg kg-1 ; Table 2) [37], only a maintenance dosage of 30 kg P2O5 ha-1 is proposed.”

Comment 6) I miss a cost/benefits analysis in this paper: is the overall proposed procedure profitable at the end? Are time and efforts reasonable with investments in proximal and remote sensing analyses?

R- The sentiment of the reviewer is natural and understandable, however, a cost / benefit analysis will require several years of data, including the costs of surveying 1 or 2 years of soil (ECa) and crop (NDVI, NDWI) variability, the costs inherent in closing the cycle (as in the case of differentiated fertilizer application that is being met in this autumn 2019 campaign). Then it will be important to evaluate the impact of the methodology on pasture productivity, quality and botanical composition, which can only be known after spring 2020.

This work is being developed as part of a research project ending in 2021, which will provide the opportunity to make a balance and cost benefit analysis.